# Renovation Strategies for Multi-Residential Buildings from the Record Years in Sweden—Profit-Driven or Socioeconomically Responsible?

**Kristina Mjörnell [1,2,]*** **, Paula Femenías [3]** **and Kerstin Annadotter [4]**

[1]   RISE Research Institutes of Sweden, 412 58 Gothenburg, Sweden
[2]   Department of Architecture and Built Environment, Lund University, 221 00 Lund, Sweden
[3]   Department of Architecture and Civil Engineering, Chalmers University of Technology, 412 96 Gothenburg, Sweden; paula.femenias@chalmers.se
[4]   KTH Royal Institute of Technology, 100 44 Stockholm, Sweden; kerstin.annadotter@abe.kth.se
*   Correspondence: kristina.mjornell@ri.se; Tel.: +46-10-516-57-45

**Abstract:** An important part of the multi-family housing stock in Sweden was built during the record years 1961–1975 and is in need of extensive renovation to be modernized. The stock is also at the center of political discussion of how to sustain 'good housing for all', especially in the rental sector. These renovation needs coincide with present energy targets and provides an opportunity to combine renovation with energy efficiency measures. Common for many of these buildings are that neglected maintenance has led to technical shortcomings, such as high energy use and low thermal comfort due to bad insulation, unsatisfactory air tightness and leaky windows, inefficient heating systems and insufficient ventilation, and moisture damage due to leaking building envelope and leaking pipes. However, the people living in these buildings are not willing to or cannot afford to pay the higher rents that extensive renovations would entail. Earlier research has highlighted the broader societal problem of energy renovations, but also that of housing companies' priority of measures with short payback times, and those that give the possibility to raise rents. However, recent observations indicated a tendency towards more holistic approaches to housing renovation, and this study was initiated to investigate how public and private housing companies deal with renovation levels, rent increases and related social problems. The main conclusions are that sustainability and social responsibility are moving up on agendas in the public sector, but also, apparently, in the renovations strategies among the private companies. What is also seen is a trend moving from extensive total renovations to more tenant-adapted and step-by-step renovations. Renovation options which do not entail such large rent increases are increasingly being seen. Implications are that housing owners favor gentle renovation with reasonable rent increases of 10%–20%, which at the same time, may be a drawback for reaching energy efficiency targets.

**Keywords:** sustainable; renovation; multi-residential buildings; affordable housing

## 1. Introduction

### 1.1. Background

Sweden, like many other nations in Europe, faces a need for large-scale urgent renovation of the post-war building stock that has passed its technical, economic, and service lifetime [1]. Neither the government nor the building sector are prepared for these challenges. There is a lack of policy regarding dealing with housing shortages and the affordability of housing. At the same time regulations are pushing for reduced energy usage, which requires investments in energy efficient renovation measures

usually involving rent increases [2]. In the last ten years, considerable research has dealt with a single focus on how to remove barriers for the successful implementation of energy-efficient renovation [3–5]. More recently, a growing number of studies have shown how energy-efficient renovation may have a negative influence with respect to environmental, economic as well as social sustainability. Evaluations using life-cycle assessments made of different renovation alternatives have shown that it is neither environmentally, socially nor economically justifiable to do extensive renovation in order to reach high energy efficiency [6,7]. Based on energy renovation projects carried out in the past, [2] estimated the cost for renovation and energy retrofitting of multi-family buildings that will reach their service life of 50 years before 2026. The authors found that the pace of renovation needs to increase but also sensitized the risk of increasing societal inequity due to increased rents as a result of deep renovation. The problem is aggravated since the buildings with extensive need of renovation are found in areas with residents from social-economic vulnerable groups. Their conclusion is that it is essential to find appropriate levels of renovation and energy efficiency measures for each building and socially sustainable payment models [2]. At the same time, studies highlighted the fact that there are often other and even more prominent motivations for renovation besides energy-efficiency, for example, urgent failure of systems and the need for improved indoor climate [8]. In fact, reports showed that few renovations that are carried out involve high ambitions for energy efficiency [9–11].

The aim of this paper was to further research current renovation strategies among housing companies. The research questions are the following: How do Swedish housing companies in the rental housing sector deal with renovation levels, rent increases as a result of renovation and the possible implications for tenants, such as displacement? Do their actions give proof of an on-going and deliberate 'low-carbon gentrification'? The scope is limited to the large Swedish stock of housing constructed during the record years and notably through the so called "Million Homes Program", a governmental housing initiative during the period of 1965–1975, subsidized both by state and municipal funding that resulted in 573,000 dwellings in multifamily buildings. Today 240,000 dwellings in this stock are owned and managed by public housing companies, about 152,000 by private companies and 181,000 by cooperative resident owned housing associations [12]. Although part of this stock has already been renovated, it is estimated that 125,000 dwellings need renovation, and stock that has been renovated before, needs new investments. Due to neglected maintenance and renovation, many building components are facing the end of their lifetime. There are also problems with insufficient indoor climate and the presence of hazardous substances. This part of stock also represents the largest potential for energy savings due to the large area of heated space [13].

Information from fourteen private and public housing companies with rental apartments, was collected in order to understand if the companies are mainly profit-driven or if they consider all sustainable aspects and specifically socially responsible renovation. The structure of the paper is first an overview of the context of sustainable renovation of housing and challenges related to energy efficiency, investments and rent increase and previous studies on housing owners' strategies for renovation. The research method is described followed by results consisting of the researchers' interpretations of fourteen housing companies' presentations at seminars. The paper ends with a discussion of the results in relation to earlier studies and finally, conclusions.

## 1.2. Literature Overview

Sustainable renovation has the potential to improve the energy performance of buildings and decreased costs for operation and maintenance, but also the housing situation and wellbeing of people with respect to indoor-climate and comfort [14], improved accessibility for elderly [15,16] and to mitigate energy poverty [17]. However, the implementation of sustainable renovation has met challenges in practice and has often led to conflicts between property owners and tenants [18,19]. In recent years, the negative implication related to increased costs for living, and in some cases, forced displacement, has been brought up [20]. Initiatives for low carbon renovation including investments in energy saving measures motivated as a means to reduce energy poverty and improve wellbeing, have

eventually led to negative implications for the residents, such as increased cost of living and in some cases, forced displacement [20]. Ecological gentrification and an 'eco-social paradox' of low carbon renovation [20] has been claimed a reality in several countries [21–24]. In reported Swedish cases, the housing companies combine technical renovation with standard improvements of the dwelling (e.g., new kitchen, extensively renovated bathrooms and wooden parquet), something which has been claimed or considered unnecessary among many tenants and done only to justify increased rents [25]. Baeten et al. [25] brought up the problem of a normalization of profit-driven renovation strategies which has been justified by what is called technical necessity and the need for carbon reductions that challenge the mission of the public housing companies to provide 'good housing for all'.

The housing stock of the record years (1961–1975) was once built as part of a welfare system. Presently, in metropolitan areas in Sweden, low income households often live in rental housing [26] and many of those live in housing areas from the Million Homes Program, where the incomes tend to be even lower than in other housing areas with rental apartments [27]. The reported Gini-coefficient which is an economic measure for income distribution, was 0.226 for Sweden the year 1991 and has increased to 0.322 the year 2017, the highest rate since the start of this measurement. A higher Gini-coefficient indicates that the development of financial standards has not been as strong for all groups, which has meant that income disparities have increased. Sweden is one of the countries where the income differences increased most from the year 2008 [28].

The increased income differences which are mainly due to lowered levels in various insurance systems, such as the sickness and unemployment funds, did not increase at the same rate as real wages, increased importance for capital income that is concentrated to the top of the distribution and demographic changes over time such as the population and household composition [29]. When looking at the measure at risk for poverty (60 percent of median income) in Sweden, 14.9 percent of the population were at risk for poverty in the year 2017. In 1991 the percentage at risk for poverty in Sweden was about seven percent [28]. The question is the following: What is the relationship between the Gini-coefficient and which households live in rental properties? It turns out that within the group with the lowest income (quartile 1), 29 percent live in owned detached houses, 16 percent live in tenant-owner association and 50 percent live in tenancy. Within the high-income group (quartile 4) 64 percent live in owned detached houses, 24 percent in tenant-owner association and 12 percent in tenancy [28].

Since the year 1991, the lowest income groups have lost more of the total income and the highest income group has gained most of the total income in Sweden. Consequently, renovation of this part of stock will be a critical issue for incumbent tenants. A report from the National Board of Housing and Planning predicts larger societal consequences of displacement as a result of deep renovation in terms of segregation [30]. The situation could be further accentuated as many municipal housing companies sell part of their housing stock to private investors to finance renovation of their remaining stock [31]. This has created a new category of owners of Million Homes program areas: private housing companies with exclusively financial interest.

Previous research has shown a lack of social ambitions in housing renovation. In many reported cases, current renovations strategies lead to rent increase beyond the tenant's ability to pay. Grossmann and Huning [22] reports on more than 7% of rent increases from energy renovations, which reportedly forced some families to leave, but also single cases with over 100% rent increase. Tirado Herrero [24] reported on a case in Vancouver where the property owner planned to raise the rents with up to 73% after renovation. These levels should be compared with studies on tenants' willingness or ability to pay for renovation measures. A Swedish study showed that more than one third of the tenants could not accept any rent increase while no more than half of the residents could accept a rent increase of 1–10 percent and very few could accept rent increases above 10 percent [32]. The Swedish Tenants' Association analyzed households' ability to pay higher rents in the city of Gothenburg [33]. They found that even a 5% rent increase would increase the amount of households in a risk group which would

not be able to pay the rent with 2%, whereas a 25% rent increase resulted in 10% more households in the risk group and a 50% rent increase would double the amount of households in the risk group.

The apprehension that companies only consider economic aspects (yield requirements and the possibility to increase rents) when choosing renovation measures has been contested in recent years. Some authors claimed that housing owners take a holistic approach to renovation and urban redevelopment where energy efficiency, along with other environmental objectives, are balanced with ambitions for social and economic development [8,34–36]. There are also studies that showed that the way that housing companies approach renovation varies depending on their size, ownership category and on their housing portfolio. Mangold [13] showed that public housing companies carry out more extensive renovation than private companies. Larger companies outperform smaller companies in terms of extensive renovation [37]. Högberg et al. [5] found a big difference between companies with a strong economic position who get instructions from their owner to reduce energy use, whereas less profitable companies with vacancies is less likely to invest in energy efficient measures. Companies with a large share of stock from the record years may not afford to do anything more than strictly profitable measures, whereas companies with a smaller share of housing from the actual period might afford to do the little extra, [5]. Nair et al. [38] found that the co-operative housing company had resource constraints that limit their effort to implement energy efficient measures, the private housing company preferred short payback times for their investments and that the public housing companies were not aware of the cost saving potential, which limited their adaption of energy-efficient measures. However, the authors emphasized that the public housing companies must implement innovative processes and products to motivate other housing companies to follow in their footsteps [38]. Lind et al. [31] analyzed the renovation strategy of a public housing company SigtunaHem in Northern Stockholm County that managed to carry out the necessary renovation measures, maintaining affordable housing through the "mini" renovation option and still demonstrated acceptable profitability, which will increase in the coming years. Based on statistics of the tenants' income (ability to pay increased rents) they offered three different renovation packages "Mini, Midi and Maxi" [31]. Finally, a recent study, reported on ten Swedish public and private housing companies' renovation strategies and how policies and objectives for energy efficiency are incorporated into their work [39]. The study clustered the housing companies into typologies distinguishing between either "commercial" or a more "societal focus" and spanning between "deep" and "partial renovation". The study revealed that few companies systematically implement deep energy renovation [39]. The large costs of deep renovation make it impossible to implement them in all renovation projects, and only public companies were found to make those investments. The situation is especially difficult for companies that have a large part of stock in the same age group and the same urgent needs for renovation. The housing companies must consider other aspects and the need to deal with social aspects and care for the tenants will conflict with costly energy renovations. As a result, most property owners in the study favored stepwise renovation spread out over time instead of all-at-once and deep renovation. The small number of cases in that study calls for further studies in order to verify these results on a larger scale.

## 2. Materials and Methods

### 2.1. Research Method

The research was qualitative and based on descriptions and statements given by fourteen housing companies about how they work with renovation of rental housing. Four seminars were held at different locations in Sweden in order to get a geographical spread and capture renovation practices in different parts of the country, covering metropolitan areas, suburbs, as well as medium-sized and smaller communities. The selection of invited companies was crucial for the validity of the study. In order to assure a good distribution of different kinds of housing companies, representatives of ten public and four private housing companies operating in the south, west, east and middle regions of Sweden were invited to the seminars. Previous studies have shown that companies acting in different economic

contexts will have varying possibilities for carrying out renovation [12]. Furthermore, the invited companies are among the major actors operating in each studied region. Another important aspect for the selection was that the companies had a long-term owner perspective on housing. We deliberately did not invite investor companies, which have a short-term interest in owning housing or housing cooperatives, consisting of owner-occupied housing. The building stock of the companies included in the study represents 16% of the apartment owned by public and 6% of the apartment owned by private multi-family housing companies in Sweden.

The companies were asked to describe their strategies, priorities, renovation practices as well as their definitions of renovations, with the purpose of answering the research questions. The requests were sent out to the representatives well in advance of the seminar. Each housing company representative had about twenty minutes at his or her disposal to shortly make an overall presentation of the company and describe their strategies, priorities, renovation practices as well as their definitions of renovations and five minutes to answer further questions from the audience and make clarifications of uncertainties. The documentation was done by audio recording of the presentations and discussions and by two researchers taking notes during the seminars. The recordings were listened to after the seminars to assure that the written notes reproduced the correct information. The housing companies shared the visual presentation and in some cases, also submitted their written manuscripts after the seminar, but this was not requested. Some companies chose not to respond to all areas.

The companies were asked to describe the following:

1.  The companies' strategies and working methods for renovation.
2.  How the companies prioritize which properties are being renovated and which measures are implemented and how.
3.  A description of the companies' different renovation levels, the companies' investment levels and rent increase principles, the companies' view of the need for technical upgrading and standard increase and the company's view of the rent regulation system.
4.  How the companies define and use the terminology, maintenance, renovation and reconstruction.

The empirical materials were analyzed using a qualitative content analysis where the researchers interpreted the expressions and concepts that the companies used and which arguments they presented. The analysis was triangulated through the involvement of three researchers in an inter-disciplinary collaboration. Furthermore, the quantitative information provided by the representatives from the companies was checked against information provided from the companies' annual reports and websites.

*2.2. Housing Companies Participating in the Study (Materials)*

The studied housing companies are of varying sizes and own buildings located in five different regions. There are both public and private housing companies and some companies also own commercial premises, offices and community properties. The companies have stocks in different age categories but all companies owned multi-residential buildings mainly built within the Million Homes program era, see Table 1.

Fictive but descriptive names were used to anonymize the companies and to avoid association with delicate information.

**Table 1.** Overview and main characteristics of the companies involved in the study.

| | Company Characteristics, Ownership | Area of Operation | Main Characteristics of Housing Stock |
|---|---|---|---|
| 1 | Public company in the suburb with a pragmatic view | Suburb to capital | 10,600 apartments, of which 80% were built during the 1960–1970s. |
| 2 | Private commercially driven subsidiary company with focus on complete renovation | Major city [1] in the south | A subsidiary company owned by a cooperative housing association. 2000 rental apartments in major city at A and B locations and about 130 apartments in the center of smaller cities but also some other facilities. |
| 3 | Public company renovating with non-displacement model | Small city [3] in major city region | One area built in the end of 1980 and the four areas built during 1960–1970s 500 apartments in total. |
| 4 | Private business-oriented company with social commitment | Suburb to major capital | Buildings built during 1970–1974 with 1300 apartments and a minimarket |
| 5 | Public company with social focus | Major city [1] in the south | 40% of the buildings were built during 1960–1970s and 40% during the 1950s. 23,500 apartments |
| 6 | Private company with relation-driven management | Major city [1] in the south | Apartments in many Swedish cities. 25,000 apartments |
| 7 | Public company with focus on solidity and life cycle perspective) | Capital | Old buildings built before 1937 as well as buildings built during 1940–1950s, 1960–1970s, 1980–1990s but also during the 2000-ties. 27,000 apartments |
| 8 | Public company with advanced portfolio management | Capital | Varying characters of buildings and year of construction mostly from 1940 to 1970. 28,000 apartments, 540,000 $m^2$ commercial space |
| 9 | Private company with focus on gentle renovation | Major city [1] in the south | Varying characters of buildings. 1600 apartments, commercial, offices and community buildings. |
| 10 | Public company with holistic approach | Small city [3] in middle region | Buildings built from 1940 to 2017, of which 900 from the 1970s. 2150 apartments, 450 homes for elderly |
| 11 | Public company with value-based portfolio management | Middle size city [2] A in middle region | 40% of the buildings built during 1965–1975s. Most apartments (6000) have low rents, low real estate value but some (200) with high rents and high real estate value |
| 12 | Public company with block wise renovation to offer different type of living | Middle size city [2] B in middle region | Most buildings are from 1960–1970s and smaller buildings from the 1800s but also new buildings. 5200 apartments, 300 commercial buildings |
| 13 | Public company with customers' focus | Middle size city [2] on major island | Many of the apartments built during the 1960s. 4600 apartments |
| 14 | Public company focusing on gentle renovation to support urban development) | Suburb to major city [1] in south west | Buildings built during 1968–1973 and renovated during 1997–2025. 2800 apartments and a few other premises |

[1] A major city is defined as having more than 200,000 inhabitants. [2] Middle size cities are defined as having 50,000–200,000 inhabitants. [3] Small cities are defined as having less than 50,000 inhabitants.

## 3. Results

The results are presented as "case" descriptions from each of the fourteen companies answering the four research questions to different extents. The descriptions are the authors' interpretation and understanding of the companies' statements.

### 3.1. Public Company in the Suburb with a Pragmatic View

The public company who acts in the suburbs of a major city, has tried different renovation concepts in different areas; one that involved complete renovation with sustainability thinking and one with limited measures. They mainly consider acute problems, security aspects, compatibility of systems, cash flow and profitability, access to internal and external resources in terms of labor, and concessions from tenants. They are well informed of the state of knowledge in the field, which means that they are confident that they implement the right solution. In recent years, they have gone from the principle of taking the opportunity to do everything at once to making less extensive measures. The renovation is financed without raising the rent too much to ensure that nobody is forced to relocate as a result of the renovation, and communication and participation in the process is important for the company. They do not throw out functioning kitchens and interior materials. They implement a minimum level of renovation, which includes renovation of the kitchen (repainting cabinet doors, new tap water mixer, lighting, new countertops), new bath and WC and new electricity. In recent years, there has been a further lower level which involves the replacement of pressurized pipes and sewage pipes, grounded outlets in bathrooms, new porcelain in bathrooms and measures for obtaining approved compulsory ventilation control. They define maintenance as measures mainly aimed at maintaining function and not increasing the value. The work is usually limited to one or a few component types and is usually handled in the yearly statement. Renovation on the other hand improves the building in some essential way and therefore, increases the value. Renovation involves many component types and the investments are handled in the balance sheet to a great extent. The company tries to avoid the term reconstruction in order not to have to fulfil the requirements for reconstruction in the building regulation. They argue that today's value-based rent setting system does not include the renovation actions that the tenants experience as most urgent to do. The company makes a lot of social efforts for children, young people, the unemployed and women, such as supporting mothers' and sisters' art projects and employing nine women full time for cleaning the property indoors and outdoors, which had led to improved perceived security in the area, decreased costs for repair and has acted as a driver for their work with sustainability.

### 3.2. Private Commercially Driven Subsidiary Company with a Focus on Complete Renovation

The subsidiary company is owned by a cooperative housing association and makes step-by-step renovation in most cases. However, their aim is to move towards total renovations including internal renovation and temporal evacuation of tenants. They have already done some pilot projects with complete renovation where the tenants were evacuated. They have rented a car going around the area to which the tenants can come and discuss. Security issues are highly prioritized by the tenants, for example, lighting. The company also works with energy projects. When it comes to finance, they have a requirement of 4% direct yield. Maintenance plans are the basis for prioritization, but they also consider reporting of deficiencies, tenant surveys and experiences reported by the accommodation host. The company let the tenant to some extent influence the internal renovation, for example, regarding the kitchen furniture, the bathroom design, and the addition of a security door. In some areas, depending on the property value, complete renovation of the apartment is considered. When the tenant terminates the contract, a vacancy inspection is made and based on that, the level of the internal renovation of the apartment is decided. The company argues that the existing rent setting model does not prioritize environmental measures and the companies do not profit from performing sustainable renovation. For example, the cost

for restoring old kitchens is not a base for increased rents, not even in protected buildings, whereas installing new kitchens induces a rent increase.

### 3.3. Public Company Renovating with a Non-Displacement Model

The public company acts in a small city, owning buildings with major renovation needs that would require measures that the company is unable to finance without increasing the rent with about 55%. A thorough survey was made to investigate the need for action for each building in the entire stock which was transferred into a maintenance system. Action plans have also been introduced to get an overview of the need for measures at a detailed level over time. The intension is to extend the life of the properties without making unnecessary efforts and to find smart energy-saving measures. The company works according to the model "Non-displacement" of existing tenants, which means that a consultation group evaluates and prepares actions with full transparency and co-determination. They have a rental model with a rent increase distributed over 10 years based on the cost principle. The intention is that the tenant should be able to remain in the apartment during and after renovation. The company has a long-term plan for their areas, including certain measures that should be taken within a seven years period and they invest approximately 22,000 Euros per apartment for ordinary renovation, including relining of pipes, changing thermostats at heat pipes, heat exchange pump on ventilation, windows and exterior maintenance and energy efficiency measures, and 50,000 Euros for total renovation. The rent increase is 1.85% per year in ten years, which adds up to approximately 20% in total. The company anticipates future government support for further renovation measures.

### 3.4. Private Business-Oriented Company with Social Commitment

The private company has a business concept based on social commitment that aims for benefit for both residents and shareholders. The company first went through different options and then set up criteria that would allow residents to stay during and after the renovation. The company has a strong local anchoring in the area, where they also have their headquarters, to have close contact with the tenants. They want to preserve the area's cultural heritage and search for ways to make use of the area's original qualities. The company wants to offer accommodation at different standard levels and start with the apartments with tenants who want renovation. They also do piece-wise renovations, which includes relining of sewage pipes, replacement of hot water circulation pipes and the renovation of bathrooms, that lead to rent increases when tenants are moving out. Their plan is to do systematic external renovation later.

### 3.5. Public Company with Social Focus

The public company makes a thorough investigation in one of the residential areas and visit every apartment to discuss and establish a relationship with each family. They experience four times higher wear in the area due to the EBO (i.e., arrange housing on your own), a legislation that gives migrants the right to arrange accommodation on their own with the help of relatives or friends. The company carries out renovation to varying degrees and invests 10 Million Euros in security measures. They have also sold out 1660 apartments to a private company to a value of 110 Million Euros. The aim for making investments in the chosen area is for one of their most vulnerable areas to be seen in a positive context. The company offers internal renovation and standard at different levels. For those who wish, they offer a slightly higher standards to keep them as residents in the area. They work a lot with collaboration between residents, school, companies and civil society organizations in the area.

### 3.6. Private Company with Relation-Driven Management

The private company acts as a security and bank for the family-owned group. Renovation is financed by the utility value principle with a rent increase of 2% per year. They have exclusively worked with step-by-step renovation without evacuation of tenants and have maintenance plans for 50 years ahead. The technical status of buildings and tenant surveys determines which renovations

should be prioritized. They work according to relationship management where they offer the residents other values such as assisting children with their homework, and involvement with the school. If the tenants like the area, they will remain in the area and this is profitable for the company, which has proven itself in several socioeconomic vulnerable areas where the damage has gone down to a minimal level. The company means that they consider 'maintenance' the aim to have high technical status over time and 'renovation' is the restoration after wear and tear, which is value increasing. 'Reconstruction' is the renewal of extensive parts. They apply a plus renovation concept when tenants are moving out that involves a complete standard improvement, including a new kitchen and new floors reaching the same standard as newly produced apartment. There is a list of measures that the tenant can choose from, and the rent increase is then negotiated with the Tenants' Association. The company argues that it is more profitable for them to increase the standard during renovation, for example, to lay wooden floors, in order to be able to increase rents. They think that the existing rent setting system is transparent but unwieldy, since each apartment needs to be negotiated separately. There are also differences between different cities; in one city, changing windows induces a rent increase while in another city, it does not. Typical yearly rent increase is 2%.

### 3.7. Public Company with a Focus on Solidity and Life Cycle Perspective

The public company acts in a major city. They base every decision on their ownership requirements and make a balance between different aspects in each project. Normally, they prioritize economic aspects, technical needs, environmental and social needs. They have an environmental policy, procurement policy and guidelines for maintenance and renovation. Usually, they do step by step renovation but sometimes, they do everything at once. They do not evacuate every tenant and they are aware that there are people who cannot stay after renovation. Technical status, complaints from tenants and social problems are reasons for renovation. The company always has a life-cycle perspective and they have a good solidity and access to capital. They think that the point of maintenance is to restore a function. The renovation of technical installations often involves the replacement of components. The renovation of building parts often means restoration to an appearance just as new. They define 're-construction' as the changing function of buildings. The company offers at least two levels of renovation where the minimum level is to secure the function of the buildings. The tenant can, to a high extent, choose to what level they want renovation. For the property owner, it is optimal with as few options as possible.

### 3.8. Public Company with Advanced Portfolio Management

The public company acts in a major city. They have a strategy for planned renovation which has been carried out on a large scale for about ten years, while previous years' renovations have been more focused on the renovation of separate components and change of installations. The priority is, above all, decided based on technical standard and age of building components. It is very costly and difficult to change the order of priority for a project, especially with regards to the evacuation of tenants and tenant influence. They define maintenance as continuous operation and maintenance measures which are budgeted based on to which extent the apartments have been renovated over the past ten years and the age of the properties. They consider major renovations as the renovation of roofs and facades, which are mostly done in the buildings from the 1940s and 1950s, where installations have already been replaced. Total reconstruction involves evacuation of tenants, internal renovation with new surfaces, new pipes, new ventilation and electricity, and in most cases, new windows and additional external insulation. The company has made an agreement with the tenants in one of their larger residential area that they are allowed to vote for three different levels of renovation at a tenants' consultation meeting. In more comprehensive renovation projects, the tenants are offered this arrangement through the local Tenant Association. The company thinks that increasing the standard of the apartments must give the tenant a feeling of value improvement. This means that it is of great importance that the tenants are given the opportunity to participate in the process and that a standard improvement and hence, the rent increase must never be speculative or unreasonably high.

### 3.9. Private Company with a Focus on Gentle Renovation

The private company works differently in different areas but argues that what gives the best return on investment is the 'low-level' internal renovations and they have showed that renovation can be done systematically with procurement at fixed prices. They make a social effort by requiring contractors to employ residents, which creates social control and less damage. Much has been invested in energy efficiency and they have saved 50% energy. They offer a customized high- or low-level renovation whereby high-level renovations usually involve relocation of the tenants. In buildings from the record years areas, they make a lot with 'low-level' renovations, which means renovating the kitchen when needed, changing bathrooms and repainting. In other areas, renovation is carried out while tenants are relocated and standard improvements are always made. The company invests 20,000 Euros for ordinary renovation and 40,000 Euros for high-level renovation and increases the rent by approximately 100 Euros per month, from 700 to 800 Euros per month, which corresponds to a 14% rent increase. They aim at long term contracts with their tenants and state that high rent increases lead to high levels of relocation of tenants, especially in low-income areas. The company thinks that the value-based rent setting system needs to be reviewed. To introduce market rents is one solution but it would not work for all tenants. The utility value rent model keeps the rent below the market rent levels, which means that the property owner subsidizes rents even for resource-intensive households.

### 3.10. Public Company with a Holistic Approach

The public company acts in a small city in the middle of Sweden. They have had high operating costs and a great need to upgrade the buildings. Now, they have modern installations in their buildings, trained staff, lower operation net cost and higher property value. The energy consumption has been reduced by 20%–30%, mainly by working with heating, lighting, additional insulation, hot water, laundry rooms and individual measurement. It has been possible to decrease the rent after renovation, and the tenants now pay individually for hot water and electricity. The company is quite small, and they do not consider that they can take a market lead in implementing new technology. They always make a trade-off between how the measures affect the economy and the social aspects and always involve the Tenant Association and their own board in decisions. Today, renovation is done step by step. First, regulatory requirements are considered and then operating costs, technical status, finances in the form of life cycle costs and rent adjustments are accounted for. The company has sold 70 apartments to finance the upgrading and maintenance of the remaining stock, which has increased the company's solidity. They define 'maintenance measures' as the measures they do to keep the original status of the building. They do not usually let the tenants choose the level of internal renovation, but when they make deeper renovation, the tenants have to choose between two levels of standard for internal renovation of their apartments. The company takes into consideration the tenants' ability to pay when they choose renovation measures and have a list of measures and related rent increases. Today, they have satisfied customers and relatively low rental levels. During renovation, tenants are evacuated to temporary modules in the courtyards.

### 3.11. Public Company with Value-Based Portfolio Management

The public company acting in a middle size city in the middle of Sweden has a well-managed stock and works with maintenance and renovation according to five-year plans. In each of their 420 buildings, it has been decided to carry out eight different measures and they have an excel sheet with an assessment scheme for needed maintenance (red, yellow, green) regarding energy use, conditions of the stairwells, laundry rooms, paved surfaces, need for external maintenance, internal maintenance and other. The goal is to not have more than 1% "red" markings, which was also fulfilled. The company holds regular meetings with the tenants and presents how they assessed the different renovation needs, in order to create consensus. They have a special plan for renovation of bathroom and have worked actively with actions for energy efficiency, renovation of roofs, insulation and district

heating. The company owns two wind turbines and is self-sufficient on electricity. They have an annual plan for sewage-pipe renovation, which accounts for the main part of their maintenance budget. When renovating the sewage pipes, they also renovate the bathrooms. They have different levels of renovation including a 'mini renovation' which, for the kitchen, would be 1. Repainting of cabinet doors. 2. Replacement of kitchen cabinets and machines. The tenants cannot individually choose renovation measures, instead, the level applied is based on an assessment made by the company. The company wants to improve the reputation in some areas without raising the standard and hence the rents and works with social issues, such as relocation, increasing the number of children that pass their grades in school and decreasing damages on premises such as scribble. In total, three employees work proactively with social issues. The typical rent increase is 10%–18%, with a maximum increase of 50 Euros per year. They give a discount on rents during renovation.

### 3.12. Public Company with Block Wise Renovation to Offer Different Type of Living

The public company acting in a small city in middle of Sweden, has a long-term strategy according to maintenance plans and manages each building block separately and create different standards in order to be able to offer different forms of accommodation. Total renovation has been carried out in one of the areas. It became very expensive and the depreciation of the value of the buildings had to be done. Now, only necessary measures are done, which include maintenance of the roofs, pipes and electricity. A plan has been made for decreasing energy use throughout the area and it is possible to halve the energy use. One of their residential areas is extremely overcrowded and has unauthorized subletting, which increases the wear and tear. In this area, it is important to have good cooperation with the tenants as well as the contractors during the renovation process. They define 'maintenance' as long-term work according to maintenance plans and 'renovation' as change of components and for example, roof renovations. The typical rent increase is by 9% after renovation, which they mean that is low compared to other municipalities nearby. They also give a rent discount during renovation.

### 3.13. Public Company with Customers' Focus

The public company acting in a middle size city has a building stock with huge renovation needs with many apartments built in the 1960s. Some energy efficiency has been carried out in the form of balancing and control measures for heating and ventilation. What initiates renovation is the end of technical life span of components and materials, or if the area is charged with social problems. There is a transition in the way of working within the company. Tenant dialogue is part of the pre-studies made for each area. The technical status, together with dialogue with the tenants, are the basis for initiating renovation projects and they let the level of the renovation be influenced by the tenants and they have the 'customer's focus instead of having the customer in focus'. They lack documented policies, guidelines and procedures for renovation but try to balance between the technical level and the financial consequences. Investments are made to create sustainable properties in the long term. The technical status is a priority, but many complaints from the tenants may lead to another order of priority. When choosing materials and solutions, sustainability and the life cycle perspectives are considered, but no systematic way of working is implemented, and no life cycle analyses are made. Extensive sales have been made to finance future renovation in other areas. The company performs renovation and standard improvements at different levels. The mini level is to repair the damage or fix failures. They try to keep the premises in good shape. The tenants can influence the standard options, for example, the choice of kitchen cabinets and the color of the flooring. Historically, they have been very cautious about raising rents during major renovations. It is optimal to find the balance between the costs of the renovation, to assure sustainability over time and offer reasonable rent levels for the tenants. The aim is to work with "mini", "midi", "maxi" solutions so as not to force any expensive renovation. However, they also get a lot of older tenants leaving their single-family house, who require higher standards, such as induction stovetop, hot air oven, bathtub and new kitchen cabinet doors. In renovation and if possible, the kitchen cabinet frames are preserved, and kitchen cabinet doors are changed to good quality. They have four service housing hosts who only work with dialogues with

residents. In one area, they have started to renovate several empty apartments and take a holistic approach, including accessibility adaptation of the bathroom, painting of the walls, renovation of staircases, and renovation of the kitchens as an option. In another housing areas, accessibility adaptation is solved with wider doors that meets the requirement for new construction. There are no increased rents directly after renovation, but when the apartment changes tenants, the rent is raised. The company increases the rent by approximately 90 Euros per year during a period of three years, which results in less than a 10% increase. They argue that when increasing the standard of the apartments, it is relatively easy to have a dialogue with the Tenants' Association regarding rent increases. However, they will start to work with rent models more systematically to avoid lengthy negotiations with the Tenants' Association in each new case. The company has problems with homeless people sleeping in the basements and has worked with sectioning of the basements, lighting, artists decorating the waste rooms and a new playground. As a social effort, they have employed students to take care of the outdoor environments during their summer break. They also offer some permanent employments to work with social activities in the housing areas and welcome all sorts of trainees, as well as people who wants to practice the language.

### 3.14. Public Company Focusing on Gentle Renovation to Support Urban Development

The public company acting in a suburb of a major city has inspected their entire building stock and stresses the importance of doing investigations of the actual status in order to take the right measures. They have house managers who are responsible for economic, ecological and social issues and have a long-term plan for action (ten years) and make special inspections of flat roofs and windows every five years. Bathrooms are refurbished if necessary and relining of the pipes is done to extend the service life. They have worked with energy efficiency, installed photovoltaics, preheated supply air via the balconies and water-saving measures. They work with gentle renovation and procure trades contracts. No rents are increased due to renovation. They have the philosophy that everyone should be able to stay in their apartments after renovation. The cost for relining of pipes is approximately 1500 Euros per apartment and 6500 Euros for the renovation of each bathroom. The company has created 3000 jobs locally and they have collaborated with five company training courses and work with supporting development of local enterprises. They arrange training together with Rotary's sailing club and football training together with one of the major football clubs in the city. The also support health promoting meeting places for the tenants and have installed a gym.

## 4. Analyses of Results and Discussion

### 4.1. Renovation Strategies and Priority of Measures

The debate has changed in Sweden during the last couple of years, from being concentrated mainly on energy efficiency to focusing more on affordable housing. A few years ago, there was a massive campaign driven by authorities and sectors organizations with the aim to fulfill the Swedish government's environmental goals and one milestone was to halve the specific energy use in the built environment until the year 2050 [40]. In order to obtain this, one mission was to convince the housing companies to make energy efficiency measures in combination with the renovation of existing buildings. A few frontrunners carried out very successful pilot projects proving that the energy use could be decreased to half with the right combination of renovation measures. The payback time was, however, very long [41]. What could also be noted was that some companies also included renovation measures raising the status of the buildings and apartments, which also gave them the possibility to increase the rent after renovation. Since the energy for heating, hot water and electricity for ventilation is usually included in the rent in Sweden, the combination of both energy efficient renovation measures as well as interior renovation measures resulted in decreased cost for the housing owner but increased rents for the tenants after renovation. Since buildings that are in most need of renovation, are in vulnerable areas, with a majority of low-income tenants [2], there is a risk that the increased rents will cause involuntary relocation.

Compared to earlier studies in the field [39], the housing companies, public as well as private, seem to move away from performing complete renovations at one occasion to gentle renovations. These total renovations have been motivated to meet high requirements on energy efficiency, or even to perform a passive house renovation. Instead, the companies' plan maintenance and renovation measures based on their tenants' income and ability to pay rents. This can be seen as a reaction to the debate of 'low carbon gentrification' [20,21,25]. As companies do less extensive renovations, with tenants remaining in the apartments, the possibilities to do energy efficiency measures are restricted to changing windows, installing heat recovery on ventilation, control and regulation of heating and ventilation and changing to energy efficient lighting.

The tenants are not involved in decisions about the renovation of installations or the external building envelope, since such investments can usually not be transferred onto the rents. However, some measures will be considered as standard raising and thus lead to a higher rent, for example, the installation of new windows and ventilation systems, which will give a better indoor climate.

All housing companies, private as well as public, bring forward the importance of keeping their tenants in the areas. From a property owner's perspective, it is economically advantageous if the tenants do not move. The housing companies seem to have met the problems with 'low carbon gentrification' by implementing strategies in which the tenants are invited to choose between optional levels of standard improvements in their apartments and related rent increases. If the tenants choose a higher standard, the rent for that specific apartment is increased. This is a strategy often used by private companies but also by some of the public ones. This model is used both when all apartments in the house are renovated and when the housing company (often a private) choose to renovate the apartments after a tenant is moving out and before a new tenant arrives, who will then pay the higher rent. One of the public companies has a strategy in which the tenants collectively can vote between three levels of standard, which is then implemented for all the apartments in the area. In any case, giving the tenants the individual option for choosing standard will result in different apartments in the same building having different standards, which the companies apparently do not see as problematic. Several of the companies, both public and private, have a low-level option in which they refurbish the original standard or only make small standard improvements, especially in low-income areas where the tenants cannot afford higher rents. However, some companies have also noticed a demand for standard-raising measures and renovate to a fully newly built standard in some areas.

There are, however, varying opinions among the companies with respect to which strategy is more economically advantageous over time, to raise the standard or to carry out low-level renovation. One public company believes that raising the standard in the apartment must give a feeling of value improvement. This means that it is important that tenants are given the opportunity to participate in the process and that the property owner knows their tenants, their needs and wishes, and economic conditions.

Some of the public companies employ a stepwise rent increase over a period of years, up to 10 years, as they are able to spread the costs of the renovation over a longer period. This will facilitate for their tenants to be able to stay in their apartment, even in the case where standard improvements are made, and rents are raised. Several of the private companies in the study are not able to spread out the investments and payback over time due to their limitation in capital and solidity. Instead, they have chosen the step-by-step model. Baeten et al. [25] argued that stepwise rent increase is also problematic as the rents will increase over time to more than the tenant can pay, and merely lead to a 'spread out the displacement' rather than offering a solution to 'low carbon gentrification'. This might be even more enforced if piecemeal renovation is used in the entire housing stock, which eventually will lead to a whole stock of housing with both high standard and rent levels.

The companies in the study seem to have a consensus that standard improvement or rent increase must never be speculative or unreasonably high. Rent increases applied among the companies in the study are typically 10%–20%, except for one public company that does not increase the rents at all. However, this is still considerably higher than tenants' willingness or ability to pay for renovation measures shown in former studies. The study by Bo-Analys [33] indicated that even with smaller

rent increases of 5%–10%, up to 25% of the current tenants might have to leave their homes. Another study showed that a third of the tenants could not accept any rent increase while nearly half of the residents could accept a rent increase of 1–10 percent and very few could accept rent increases above ten percent [32].

### 4.2. The Companies' Views on the Current Rent Setting System

There are various opinions regarding the question of whether the "renovation debt" is to be paid by the tenants, the housing companies, the municipality or the state. Some companies claimed that the tenants did not pay for maintenance and wear whereas the Tenants' Association argues that the tenants have already paid for maintenance and wear from their rents over the years. The construction of the tax system gives no incentives for the rental housing companies, private or public to build up rent funds as they would be taxed 25% each year. This is because rents over time must cover all costs for maintenance and wear so that renovation can be financed by renovation funds. The tenants have not paid for the wear and tear during the years they have lived there, and since the company have difficulties invoicing retroactively, the "maintenance debt" must be amortized in the future. The system therefore forces them to implement value-added measures even though such measures are not always necessary.

The rent setting system does not seem to be equal in each location, even though this needs to be discussed with the Swedish Tenants' Association. The association for public housing companies has developed a guide for systematic rent setting to clarify what measures are affecting the rent increase but also subsidies for renovation. One private company gives the example that in one municipality, changing a window is a basis for a rent increase, whereas in another, it is not. The companies have been given different possibilities for rent increase, where one company has given the right to increase the rent based on maintenance, whereas others did not even know that it was a possibility to initiate negotiations about rent levels. One company argued that the value-based rent model does not consider the most important aspects according to the tenants, such as cultural historical values and indoor environmental aspects. Another company questions whether it is decent to rent out apartments with droughty windows, bubbly plastic flooring and high energy consumption. It depends on many factors, where the situation, the price and your other options for housing play a major role.

From an environmental point of view, it is problematic that rents, which is the basic income for a housing company, can only be raised if the utility value, i.e., the standard, is raised [31]. Today's renting system thus makes it impossible to renovate to a level that only restores the existing standard. Today's renting system does encourage renovation measures which merely restore the property to the existing standard, since such restoration measures always require an investment but do not induce any rent increase. One of the private companies gives the example regarding the misconception that it is possible to save, for example, kitchens and interior doors in order to minimize costs. In the current economic and rent setting system, it will, on the one hand, in fact, be very expensive to take out, store, move back and handle all materials during the renovation. On the other hand, the refurbished or restored interior is not considered as an improvement of the standard and thus does not entitle any rent increase and gives no incentives to the housing companies to make those kinds of efforts.

### 4.3. The Companies' Use of Nomenclature

The companies seem to use the possibility of applying different nomenclatures as a mean to allocate costs in renovation, and large differences are found in how they nominate renovation. Maintenance measures are included in the income statement, while renovation measures are included in the balance sheet. Some companies avoid cost-driving requirements of 'reconstruction' (normally, new building regulations apply to reconstruction such as full accessibility for physically impaired persons, etc.) by instead terming the measures maintenance and renovation. One company makes a difference between maintenance and restoration of function, and renovation, which they define as measures significantly improving the building and thus increases the value. Another company defines maintenance as restoring function, renovation as restoring to new condition while they define

reconstruction as changing function. Yet another company refers to value-enhancing measures as renovation while another company refers to maintenance as ongoing measures that are budgeted annually and by renovation, they mean larger measures such as roofs and facades and they use the expression reconstruction only for complete renovations both including measures internally and externally while tenants are evacuated.

*4.4. Representativity of Results*

Finally, it could be argued that more companies could have been included in the study. However, the companies participating in the study owned 16% of the apartments in public housing companies and 6% in private housing companies in Sweden. However, the results show a consistency in how renovation is dealt with in different regions, smaller and bigger cities as well as suburbs and for different sizes of companies to reflect an on-going trend. What could be jeopardizing the validity of the study is rather that renovation practices are sensitive to changes in public opinion, dependent on political priorities as well as technological advances and economic prerequisites and studies of renovation practices should be carried out with frequency.

## 5. Conclusions

This study includes ten public housing companies and four private housing companies in Sweden. In all the public housing companies and one of the private housing companies, a major part of their buildings was constructed during the record years 1961–1975.

Conclusions that can be drawn from the companies' self-reported strategies for renovation of their properties are firstly that a majority, twelve of the fourteen companies, are focusing on basic renovation measures. The companies have given priority to technical function and extending the lifetime of buildings such as the renovation of roofs, changing windows, changing or relining of pipes, accessibility adaption and electricity measures. Interestingly, two of the companies have invested in their own energy sources in form of wind and solar power of which the wind turbines make the company self-sufficient on electricity.

We conclude that the decisions the companies have made regarding the basic renovation strategies are informed decisions based on their knowledge of the tenants' economic situations. Rent increases after renovations applied among the companies in this study are about 10%–20%, which are considered relatively low compared to strategies involving deep renovation. However, earlier studies of tenants' willingness to pay higher rents after renovation have shown that in general, more than half of the tenants could not accept such increases. Our conclusion is that the difference between what tenants can pay for and how much the rent is increased after renovations do not match, which can cause problems of displacement for several tenants within years to come. The renovation strategy "Mini, Midi, Maxi" has inspired most of the companies in the study and can be the efficient remedy against displacement as it allows the tenants to choose renovation level and associated rent increase for their own apartment. Some companies are also experimenting with a so-called "0-level" renovation, involving limited intervention and no rent increase. Financing of renovations is a huge problem for the public companies and without access to government loans with low interest rate in sight, it seems like their only opportunity is to sell parts of the housing stock to private companies. The questions remain what the new owners will do with the purchased stock. If they start extensive renovation without involving tenants, there may be a risk of gentrification and displacement regarding the tenants in the sold housing stock.

The social sustainability efforts made by the companies prove that they have developed a deeper understanding of the value of social sustainability work. Employed persons working for the housing company are often from the residential area. Cleaning insides and outsides bring other important results than a clean and proper environment, such as perceived security, lower costs for repair and recognizing people in the area. Most of the companies work with social efforts for children, young people and civil society organizations in the residential area. Our conclusion is that measures within

social sustainability are important for inclusion in the residential area, the community and the society. The results of this study indicate that focus on affordability and lowering the costs for renovation also has implications for energy efficiency. To reach high levels of energy efficiency, extensive renovation is required, which, in most cases, involves additional insulation of the building envelope and new heating and ventilation systems and this often implies costly temporal relocation of tenants during renovation work. With the current trend towards more moderate renovations, it is not possible to reach the European or national ambitious goals for energy efficient renovation. The conclusion is that energy policy, to a larger extent, needs to be aligned with housing policy in order to reach socio-economic responsible and sustainable renovation.

**Author Contributions:** Conceptualization, K.M., P.F. and K.A.; methodology, K.M., P.F. and K.A.; validation, K.M., P.F. and K.A.; formal analysis, K.M., P.F. and K.A.; investigation, K.M., K.A. and P.F.; resources, K.M., K.A. and P.F.; data curation, K.M., K.A. and P.F.; writing—original draft preparation, K.M., P.F. and K.A.; writing—review and editing, K.M., P.F. and K.A.; project administration, K.M.; funding acquisition, K.M. and P.F.

**Funding:** The research was covered by the grant number 2013–1804 SIRen, the national research environment on sustainable integrated renovation, funded by the Swedish Research Council Formas.

**Acknowledgments:** We thank the representatives of the fourteen housing companies for sharing information of their renovation strategies end experience from implementing them in real renovation projects. We also thank the reviewers and editors for providing constructive feedback on the manuscript.

**Conflicts of Interest:** The authors declare no conflict of interest.

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
