# Peer review of "Renovation Strategies for Multi-Residential Buildings from the Record Years in Sweden—Profit-Driven or Socioeconomically Responsible?"

_sustainability, doi:10.3390/su11246988_

Round 1

Reviewer 1 Report

Brief summary - The aim of the work is to deepen the understanding of current renovation strategies used by public and private owners of rental housing (specifically, multi-residential buildings from the Record Years in Sweden), based on the consideration of societal problem of energy renovations and the subsequent implications in terms of increased rents for tenants.

Broad comments - Overall, the article is well written, organized and balanced in its structure and style, although the Bibliographic reference section is a little too limited. The topic is a very current one and certainly falls within one major concern areas of sustainable building development. The chosen methodology of the study – seminar and interviews - is also original and very interesting.

Nevertheless, the work has in its methodological setting also some points of weakness. As it is substantially based on statements from the interviewees, the method requires some form of control for answers’ reliability, but no measure seems to have been implemented for this issue. Furthermore, it presents more than one heterogeneity factor, such as differences in the allowed scheduling (some firms have been allowed to send answers after the seminars) and the long time span for the whole study (two years), with the risk of having some of the answers influenced by changed conjunctures and context conditions and, as such, not fully comparable to the others. Another limitation is the unavailability of the full list of questions (sub-questions are not reported). The definitions chosen for identifying each firm could be better formulated: for example, the company in par.3.10 is described as “Public company with holistic approach with tenant involvement”, while also many other firms described include forms of tenant involvement. It is also not so clear if the same definition really corresponds to an actually performed involvement, judging from lines 378-380.

Issues pertaining to the conflict between renovation strategies and the fulfilment of energy targets are not adequately addressed, since the energy-related matters in themselves are insufficiently investigated. The work focuses on the relation between the type of intervention and the related rent increase, and on the response of tenants to that. The expression ‘energy efficiency’ can be found only a couple of times in the whole Section 3 (3.9, 3.11, 3.12), while the relation between rent costs and the yearly/monthly financial savings obtainable through energy saving measures should be extensively investigated in the study. The article’s main flaw, in my opinion, is that energy savings related to energy efficient measures are left out of the picture in a dissertation that wants to focus on housing affordability; the use of renewable energies and energy self-sufficiency (apart from par.3.14) should also be dealt with more thoroughly. Unfortunately, these topics are not addressed in the formulation of the seminars’ questions.

The ‘Conclusions’ section is somehow contradictory in its first part (lines 591-597): the conclusion that companies have made informed decisions based on the knowledge of the tenants’ economic situations is in conflict with the conclusion that “the difference between what tenants can pay for and how much the rent is increased after renovations does not match”: this part should be better formulated.

The article’s aim and topic are really of great relevance for the intended Special Issue ("Sustainable Built Environment and Future Proof Innovations"); the basic planning of the proposed research could really add to the current knowledge on the theme, but it needs to be better focused and presented.

Specific comments - Overall, the English language is fine, with just some grammatical imprecision, especially in the Section 3. A check of singular vs. plural verbs used in relation to the subject is suggested.

Line 277: “…tenants who wants…”

Line 282: “the public company makes”

Line 284: “a legislation that give”

Line 289: “The company offers” vs. Line 320: “The company always have” vs. Lines 433-434: “The company performs”

Line 310: “each apartment need”

Line 311: “changing windows induce”

Lines 378-380: something is missing: re-phrase the sentence

Lines 428-429: “…try to balance between the technical level in relation to…”: re-phrase as "between…. and..."

Line 429: “Investments is made”

Lines 437-438: “It is optimal to find the balance between the cost of renovation”

Line 439: “They aim is”

Line 472: “together with a one of the major…”

Line 486: “All housing companies, private as well as public they bring…”

Lines 492-493: “This model is employed in both the case that all apartments are renovated or in the case that…”

Lines 542-543: “…in order to make it clear what kind of measures that are basis for…”

Line 528: “settting”

Line 607: “Cleaning inside and outside bring other…”

Author Response

Dear Sirs,

First of all, we would like to thank the reviewers for their comments and recommendations, which have been very useful for clarification and improvement of the paper. We have considered most of the comments and added text explaining the method in more detail, even thought, the study as such could not be changed.

We hope that you are satisfied with our amendments and that the paper could be published.

Kind regards,

Kristina, Paula and Kerstin

Reviewer 1

Broad comments –

Comment 1: Overall, the article is well written, organized and balanced in its structure and style, although the Bibliographic reference section is a little too limited.

Answer: The literature review has been extended with a few references.

Sustainable renovation has the potential of improving the housing situation and wellbeing for people, with respect to indoor-climate and comfort (Mjörnell et al. 2010) and improved accessibility for elderly (Kovacic et al. 2015; Serrano-Jimenez et al, 2018) or to mitigate energy poverty (Simcock 2017.) However, the implementation of sustainable renovation has met challenges in practice and has often led to conflicts between property owners and tenants (Hoppe 2012; Astmarsson et al. 2013). In later years the negative implication related to increased costs for living, and in some cases forced displacement has been brought up [14].

Mjörnell, K., Kovacs, P., Hägerhed Engman, L., Gustavsson, T., and Ylmen, P. (2010). Monitoring of indoor environment and energy use in the renovated buildings on Brogården in Alingsås. Proceedings of Passive house conference 2010. Ástmarsson, B., Jensen, P. A., & Maslesa, E. (2013). Sustainable renovation of residential buildings and the landlord/tenant dilemma. Energy Policy, 63(0), 355–362. https://doi.org/10.1016/j.enpol.2013.08.046 Hoppe, T. (2012). Adoption of innovative energy systems in social housing: Lessons from eight large-scale renovation projects in The Netherlands. Energy Policy, 51, 791–801. Kovacic, I., Summer, M., & Achammer, C. (2015). Strategies of building stock renovation for ageing society. Journal of Cleaner Production, 88, 349–357. Serrano-Jiménez, A., Barrios-Padura, Á., & Molina-Huelva, M. (2018). Sustainable building renovation for an ageing population: Decision support system through an integral assessment method of architectural interventions. Sustainable Cities and Society, 39, 144–154. Simcock, N., Thomson, H., Petrova, S. and Bouzarovski, S. Energy Poverty and Vulnerability: A Global Perspective. 2017. Farsäter, K., Wahlström, Å., Mjörnell, K., & Johansson, D. (2015). A synthesis of studies on renovation profitability. 9-18.

Comment 2:

The topic is a very current one and certainly falls within one major concern areas of sustainable building development. The chosen methodology of the study – seminar and interviews - is also original and very interesting.

Nevertheless, the work has in its methodological setting also some points of weakness. As it is substantially based on statements from the interviewees, the method requires some form of control for answers’ reliability, but no measure seems to have been implemented for this issue.

Answer:

The method has been described in more detail. The control of the answers’ reliability was that at least two researchers listen, made notes and the seminars were recorded. Intended lies could of course not be identified even though unreasonable answers could be questioned during the seminar. Then text has been changed to:

The research is qualitative and based on descriptions and statements given by fourteen housing companies about how they work with renovation of rental housing. Four seminars were held at different locations in Sweden in order to get a geographical spread and capture renovation practices in different parts of the country, covering metropolitan areas, suburbs, as well as medium-sized and smaller communities. The selection of invited companies will be crucial for the validity of the study. In order to assure a good distribution of different kinds of housing companies, representatives of ten public and four private housing companies operating in the regions were invited to the seminars. Previous studies have shown that companies acting in different economic contexts will have varying possibilities for carrying out renovation [12]. Furthermore, the invited companies are among the major actors operating in each studied region. Another important aspect for the selection was that the companies had a long-term owner perspective of housing. We have deliberately not invited investor companies, which have a short-term interest in owning housing or housing cooperatives, consisting of owner-occupied housing. The building stock of the companies included in the study represents 16% of the apartment owned by public and 6% of the apartment owned by private multi-family housing companies.

The companies were asked to describe their strategies, priorities, renovation practices as well as their definitions of renovations, with the purpose of answering the research questions. The requests were sent out to the representatives well in advance of the seminar. Each housing company representative had about twenty minutes at his or her disposal to shortly make an overall presentation of the company and describe their strategies, priorities, renovation practices as well as their definitions of renovations and five minutes to answer further questions from the audience and make clarifications of uncertainties. The documentation was done by audio recording of the presentations and discussions and by two researchers taking notes during the seminars. The recordings were listened through after the seminars to assure that the written notes reproduced the correct information. The housing companies have shared the visual presentation and, in some cases, also submitted their written manuscripts after the seminar. Some companies have chosen not to respond to all areas.

The companies were asked to describe the following:

The companies' strategies and working methods for renovation. How the companies prioritize which properties are being renovated and which measures are implemented and how. A description of the companies' different renovation levels, the companies' investment levels and rent increase principles, the companies' view of the need for technical upgrading and standard increase and the company's view of the rent regulation system. How the companies define and use the terminology; maintenance, renovation and reconstruction. The empirical materials has been analyzed using qualitative content analysis where the researchers have interpreted which expressions and concepts that the companies have used, which arguments they presented, if the arguments can be derived from the companies' core values ​​and goals, and in which way they relate to political goals and society's laws and regulations. The analysis has been triangulated through the involvement of three researchers in an inter-disciplinary collaboration. Furthermore, the quantitative information provided by the representatives from the companies has been checked against information provided from the companies’ annual reports and websites.

Comment:

Furthermore, it presents more than one heterogeneity factor, such as differences in the allowed scheduling (some firms have been allowed to send answers after the seminars) and the long time span for the whole study (two years), with the risk of having some of the answers influenced by changed conjunctures and context conditions and, as such, not fully comparable to the others.

Answer:

The seminars were arranged during the course of two years for practical reasons. This is of course a drawback as some conditions change in the outside world, however renovation strategies are long term strategies and are not subjected to major changes during the period of two years. The documentation sent in by the companies consisted of the power point file and the manuscripts for their presentations, which they shared with the researchers.

Comment:

Another limitation is the unavailability of the full list of questions (sub-questions are not reported).

Answer:

We used open questions and the housing companies were rather free to present their strategies, priorities, renovation practices as well as their definitions of renovations and the researchers and the researchers did not force them to strictly follow the questions.

Comment:

The definitions chosen for identifying each firm could be better formulated: for example, the company in par.3.10 is described as “Public company with holistic approach with tenant involvement”, while also many other firms described include forms of tenant involvement. It is also not so clear if the same definition really corresponds to an actually performed involvement, judging from lines 378-380.

Answer:

The “nicknames” were invented to avoid using the real names of the companies but something more descriptive than company A, B, C… The last part of the name “with tenant involvement” has been removed.

Comment:

Issues pertaining to the conflict between renovation strategies and the fulfilment of energy targets are not adequately addressed, since the energy-related matters in themselves are insufficiently investigated. The work focuses on the relation between the type of intervention and the related rent increase, and on the response of tenants to that. The expression ‘energy efficiency’ can be found only a couple of times in the whole Section 3 (3.9, 3.11, 3.12), while the relation between rent costs and the yearly/monthly financial savings obtainable through energy saving measures should be extensively investigated in the study. The article’s main flaw, in my opinion, is that energy savings related to energy efficient measures are left out of the picture in a dissertation that wants to focus on housing affordability; the use of renewable energies and energy self-sufficiency (apart from par.3.14) should also be dealt with more thoroughly. Unfortunately, these topics are not addressed in the formulation of the seminars’ questions.

Answer:

The following text is added to section 4.1.

The debate has changed in Sweden during the last couple of years, from being concentrated mainly on energy efficiency to focus more on affordable housing. A few years ago, there was a massive campaign driven by authorities and sectors organizations with the aim to fulfill the environmental goals and to decrease emissions from all sectors. One mission was to convince the housing companies to make energy efficient renovation. A few frontrunners carried out very successful pilot projects proving that the energy use could be decreased to half with the right combination of renovation measures. The payback time was however very long (Farsäter et al. 2015). What could also be noted was that some companies also included renovation measures raising the status of the buildings and apartments, which also gave them the possibility to increase the rent after renovation. Since the energy for heating, hot water and electricity for ventilation is usually included in the rent in Sweden, the combination of both energy efficient renovation measures as well as interior renovation measures resulted in decreased cost for the housing owner but increased rents for the tenants after renovation. Since the building stock in most need of renovation is located in vulnerable areas, with a majority of low-income tenants, the increasing rents could cause involuntary relocation.

Comment:

The ‘Conclusions’ section is somehow contradictory in its first part (lines 591-597): the conclusion that companies have made informed decisions based on the knowledge of the tenants’ economic situations is in conflict with the conclusion that “the difference between what tenants can pay for and how much the rent is increased after renovations does not match”: this part should be better formulated.

Answer:

The text has been slightly changed to:

We conclude that the decisions the companies have made regarding the basic renovation strategies are informed decisions based on their knowledge of the tenants' economic situations. Rent increases after renovations applied among the companies in this study are about 10-20%, which are considered relatively low compared to strategies involving deep renovation. However, earlier studies of tenants’ willingness to pay higher rents has shown that in general, more than half of the tenants could not accept such increases.

Comment:

The article’s aim and topic are really of great relevance for the intended Special Issue ("Sustainable Built Environment and Future Proof Innovations"); the basic planning of the proposed research could really add to the current knowledge on the theme, but it needs to be better focused and presented.

Answer:

We think that this article is n important contribution to show the change of housing companies’ renovation strategies towards more gentle renovation with limited or no rent increase. The study was carried out a couple of years ago within a research environment, and it is not possible to change the method the study was performed. If the study is repeated in a couple of years, improvements of the method and research questions could be valuable to take into account. 

Answers to Specific comments - Overall, the English language is fine, with just some grammatical imprecision, especially in the Section 3. A check of singular vs. plural verbs used in relation to the subject is suggested.

The following has been corrected:

Line 277: “…tenants who wants…” corrected to “...tenants who want”

Line 282: “the public company makes” corrected to “the public company make”

Line 284: “a legislation that give” corrected to “a legislation that gives”

Line 289: “The company offers” vs. corrected to “the company offer”

Line 320: “The company always have” vs. OK

Lines 433-434: “The company performs” corrected to “the company perform”

Line 310: “each apartment need” corrected to “each apartment needs to”

Line 311: “changing windows induce” corrected to “changing windows induces”

Lines 378-380: something is missing: re-phrase the sentence The sentence was rephrased to: The company has sold 70 apartments to finance the upgrading and maintenance of the remaining stock which has increased the company's solidity. They define ‘maintenance measures’ as the measures they do to keep the original status of the building. They do not usually let the tenants choose the level of internal renovation, but when they make deeper renovation, the tenants have to choose between two levels of standard for internal renovation of their apartments.

Lines 428-429: “…try to balance between the technical level in relation to…”: re-phrase as "between…. and..." Corrected to: balance between the technical level and the financial consequences.

Line 429: “Investments is made” corrected to “investments are”

Lines 437-438: “It is optimal to find the balance between the cost of renovation” Corrected to “costs”

Line 439: “They aim is” Corrected to “The aim is”

Line 472: “together with a one of the major…” corrected to: “together with one of the major”

Line 486: “All housing companies, private as well as public they bring…” Corrected to: All housing companies, private as well as public bring

Lines 492-493: “This model is employed in both the case that all apartments are renovated or in the case that…” Corrected to: This model is used both when all apartments in the house are renovated and when the housing company (often a private) choose to renovate the apartments after a tenant is moving out and before a new tenant arrives, who will then pay the higher rent.

Lines 542-543: “…in order to make it clear what kind of measures that are basis for…” Corrected to: The association for public housing companies has developed a guide for systematic rent setting to clarify what measures are affecting the rent increase but also subsidies for renovation.

Line 528: “settting” Corrected to “setting”

Line 607: “Cleaning inside and outside bring other…” Corrected to: Cleaning insides and outsides bring other important results than a clean and proper environment,

Reviewer 2 Report

General remarks

The authors investigate the renovation strategies and rent policies of Swedish landlords. The authors briefly highlight that rent increases following profit-driven renovation strategies are a global phenomenon. While profit-driven renovation strategies can be found in other countries, the authors could explain more thoroughly why the results of their research are relevant to an international audience. The research result found very little proof to support claims of "low carbon gentrification" in the Swedish rental housing sector.

The authors use the development of the GINI-coefficient to illustrate increasing inequality in Sweden (line 96-98), but do not clarify how housing costs have affected this development. It would be fair to mention that Sweden is still in the top-10 countries with the lowest GINI-coefficient. 

The authors predominantly focus on rent increases but say very little on the impact of energy retrofits on lower energy costs. It would make the paper more balanced if it would also take into account the cost-benefits for residents.

The authors mention that their paper aims to research current renovation strategies among housing companies (line 60), but it remains unclear what the main research question is.  

Methodology

The authors use the presentations of 14 housing companies as the empirical core of their paper. There are some unclarities in the applied methodology. For example, how were the 14 cases selected? What share of the rental sector do these companies represent?

The authors should explain more in-depth how they analysed the presentations of the 14 companies and how they connected this data to the "core values and goals, political goals and society's laws and regulations" (lines 185-186). Which documents did the authors use for this analysis?  

The authors base their research results on their "interpretations and understanding of the companies' presentations". Was this in any way or form validated by the companies? 

The authors appear to triangulate statements of the companies by investigating documents (for example, on line 567-568 income statements and balance sheets are mentioned). Is this anecdotal, or did the authors validate the claims of the 14 companies in a more structured way? 

Author Response

Answers to Review comments

 Dear Sirs,

 First of all, we would like to thank the reviewers for their comments and recommendations, which have been very useful for clarification and improvement of the paper. We have considered most of the comments and added text explaining the method in more detail, even thought, the study as such could not be changed.

We hope that you are satisfied with our amendments and that the paper could be published.

 Kind regards,

Kristina, Paula and Kerstin

Reviewer 2

General remarks

Remark:

The authors investigate the renovation strategies and rent policies of Swedish landlords. The authors briefly highlight that rent increases following profit-driven renovation strategies are a global phenomenon. While profit-driven renovation strategies can be found in other countries, the authors could explain more thoroughly why the results of their research are relevant to an international audience. The research result found very little proof to support claims of "low carbon gentrification" in the Swedish rental housing sector.

Answer:

Text was added in section 4.1:

The debate has changed in Sweden during the last couple of years, from being concentrated mainly on energy efficiency to focus more on affordable housing. A few years ago, there was a massive campaign driven by authorities and sectors organizations with the aim to fulfill the environmental goals and to decrease emissions from all sectors. One mission was to convince the housing companies to make energy efficient renovation. A few frontrunners carried out very successful pilot projects proving that the energy use could be decreased to half with the right combination of renovation measures. The payback time was however very long (Farsäter et al. 2015). What could also be noted was that some companies also included renovation measures raising the status of the buildings and apartments, which also gave them the possibility to increase the rent after renovation. Since the energy for heating, hot water and electricity for ventilation is usually included in the rent in Sweden, the combination of both energy efficient renovation measures as well as interior renovation measures resulted in decreased cost for the housing owner but increased rents for the tenants after renovation. Since the building stock in most need of renovation is located in vulnerable areas, with a majority of low-income tenants, the increasing rents could cause involuntary relocation.

 Remark:

The authors use the development of the GINI-coefficient to illustrate increasing inequality in Sweden (line 96-98), but do not clarify how housing costs have affected this development. It would be fair to mention that Sweden is still in the top-10 countries with the lowest GINI-coefficient. 

Answer:

Explaining text and two references has been added:

The reported Gini-coefficient which is an economic measure for income distribution, was 0.226 for Sweden the year 1991 and has increased to 0.322 the year 2017, the highest rate since the start of this measurement. A higher Gini-coefficient indicates that the development of financial standards has not been as strong for all groups, which has meant that income disparities have increased. Sweden is one of the countries where the income differences increased most from the year 2008 [28].

The increased income differences which are mainly due to: lowered levels in various insurance systems such as the sickness and unemployment funds did not increase at the same rate as real wages, increased importance for capital income that is concentrated to the top of the distribution and demographic changes over time such as the population and household composition [29]. When looking at the measure at risk for poverty (60 percent of median income) in Sweden, 14.9 percent of the population are at risk for poverty in the year 2017. In 1991 the percent at risk for poverty in Sweden was about 7 percent [28]. The question is: What is the relationship between the Gini-coefficient and which households live in rental properties? It turns out that within the group with the lowest income (quartile 1), 29 percent live in owned detached houses, 16 percent live in tenant-owner association and 50 percent live in tenancy. Within the high-income group (quartile 4) 64 percent lives in owned detached houses, 24 percent in tenant-owner association and 12 percent in tenancy [28].

Remark:

The authors predominantly focus on rent increases but say very little on the impact of energy retrofits on lower energy costs. It would make the paper more balanced if it would also take into account the cost-benefits for residents.

Answer:

See comment above about the energy costs included in the rents.

Remark:

The authors mention that their paper aims to research current renovation strategies among housing companies (line 60), but it remains unclear what the main research question is.  

Answer:

The aim and research questions have been clarified and the following text:

The aim of this paper is to further research current renovation strategies among housing companies. The research questions are: How do Swedish housing companies in the rental housing sector deal with renovation levels, rent increases as a result of renovation and the possible implications for tenants, such as displacement? Do their actions give proof of an on-going and deliberate ‘low-carbon gentrification’?

Methodology

Comment: The authors use the presentations of 14 housing companies as the empirical core of their paper. There are some unclarities in the applied methodology. For example, how were the 14 cases selected? What share of the rental sector do these companies represent?

Answer:

The selection of invited companies will be crucial for the validity of the study. In order to assure a good distribution of different kinds of housing companies, representatives of ten public and four private housing companies operating in the regions were invited to the seminars. Previous studies have shown that companies acting in different economic contexts will have varying possibilities for carrying out renovation [12]. Furthermore, the invited companies are among the major actors operating in each studied region. Another important aspect for the selection was that the companies had a long-term owner perspective of housing. We have deliberately not invited investor companies, which have a short-term interest in owning housing or housing cooperatives, consisting of owner-occupied housing.

 The building stock of the companies included in the study represents 16% of public and 6% of private owned multi-family housing, in terms of numbers of apartments.

Comment:

The authors should explain more in-depth how they analysed the presentations of the 14 companies and how they connected this data to the "core values and goals, political goals and society's laws and regulations" (lines 185-186). Which documents did the authors use for this analysis? 

Answer:

The text has been changed to:

The empirical materials have been analyzed using qualitative content analysis where the researchers have interpreted which expressions and concepts that the companies have used and which arguments they presented. The analysis has been triangulated through the involvement of three researchers in an inter-disciplinary collaboration. Furthermore, the quantitative information provided by the representatives from the companies has been checked against information provided from the companies’ annual reports and websites.

Comment:

The authors base their research results on their "interpretations and understanding of the companies' presentations". Was this in any way or form validated by the companies? 

Answer:

The following text was added: The documentation was done by audio recording of the presentations and discussions and by two researchers taking notes during the seminars. The recordings were listened through after the seminars to assure that the written notes reproduced the correct information.

Comment:

The authors appear to triangulate statements of the companies by investigating documents (for example, on line 567-568 income statements and balance sheets are mentioned). Is this anecdotal, or did the authors validate the claims of the 14 companies in a more structured way? 

Answer:

No, this is only mentioned as an example of how they report different renovation measure in the financial reports.

Other changes:

The abstract was made more distinct by adding the text:

However, recent observations indicate a tendency towards more holistic approaches to housing renovation, and this study was initiated to investigate how public and private housing companies deal with renovation levels, rent increases and related social problems.

Reviewer 3 Report

The paper contains the results of qualitative analysis of 14 housing companies in Sweden about how they work with renovation of rental housing. Review of previous researches is well written. Descriptions of renovation strategies for 14 housing companies are also well described. It would be much better if authors describe whether there is representativeness of selecting 14 companies among other housing companies throughout the nation. Analysis itself is okay. However, relationship between the content of the paper and sustainability is not clear enough. I am not sure whether the paper draw attention of the reader of the journal. Authors should consider submitting to other journals focusing on building issues.

Author Response

Answers to Review comments

 Dear Sirs,

 First of all, we would like to thank the reviewers for their comments and recommendations, which have been very useful for clarification and improvement of the paper. We have considered most of the comments and added text explaining the method in more detail, even thought, the study as such could not be changed.

We hope that you are satisfied with our amendments and that the paper could be published.

 Kind regards,

Kristina, Paula and Kerstin

Reviewer 3

Comment:

The paper contains the results of qualitative analysis of 14 housing companies in Sweden about how they work with renovation of rental housing. Review of previous researches is well written. Descriptions of renovation strategies for 14 housing companies are also well described. It would be much better if authors describe whether there is representativeness of selecting 14 companies among other housing companies throughout the nation. Analysis itself is okay. However, relationship between the content of the paper and sustainability is not clear enough. I am not sure whether the paper draw attention of the reader of the journal. Authors should consider submitting to other journals focusing on building issues.

Answer:

We are convinced that the paper fits well to the journal, encouraged by one of the reviewers comment: The article’s aim and topic are really of great relevance for the intended Special Issue ("Sustainable Built Environment and Future Proof Innovations"); the basic planning of the proposed research could really add to the current knowledge on the theme.

The authors have published a couple of articles in the journal before and are frequent readers and reviewers of other articles published in the journal.

The manuscript has been revised according to the two other reviewers’ comments.

The abstract was made more distinct by adding the text:

However, recent observations indicate a tendency towards more holistic approaches to housing renovation, and this study was initiated to investigate how public and private housing companies deal with renovation levels, rent increases and related social problems.

Round 2

Reviewer 1 Report

Apart from the proposed grammatical corrections, the revised version of the article addresses almost all my comments, giving a better explanation of unclear points.

Unfortunately, its main limitation - i.e. the exclusion of considerations about life-cycle costs and savings, and about renewable energy strategies - was impossible to solve, because those points should have been included in the preliminary design of the research. So, my overall rating of the works remains "Average" with respect to the Special Issue scope.

Since the article makes a positive contribution to show the change in companies' trends and attitudes towards "soft" low-increase renovation strategies, it is anyway an interesting point of view to be included in the Special Issue, on condition that the exclusion of life-cycle savings in energy-efficient strategies is somehow explicitly mentioned in the text and a reason for that is given - at least, a couple of lines.

For the future prosecution of this research, I warmly suggest the authors including those aspects in their investigations, with which the work will achieve a higher profile in terms of completeness and scientific soundness.

Author Response

Dear Sirs,

If I understand it right, the reviewer is satisfied with the changes made in the manuscript. We thank you ones again for the comments which supported us to improve the paper. In other related research projects we have thoroughly studied the environmental impact from renovation measures and the work has been published in Sustainability, see reference below, but this was not the scope of this paper. 

Kind regards,

Kristina Mjörnell

Malmgren L., Mjörnell, K., Application of a Decision Support Tool in Three Renovation Projects September 2015 Sustainability Sustainability(7):12521-12538 DOI: 10.3390/su70912521

Reviewer 3 Report

The revised paper is much better than the original paper. Representativeness of selecting 14 companies among other housing companies throughout the nation is described and limitations of the research are well written.

Author Response

Dear Sirs,

If I understand it right, the reviewer is satisfied with the changes made in the manuscript. We thank you ones again for the comments which supported us to improve the paper.

Kind regards,

Kristina Mjörnell